

# Scaling-up Permafrost Thermal Measurements in Western Alaska using an Ecotype Approach

William L. Cable[1], Vladimir E. Romanovsky[1], M. Torre Jorgenson[2]

[1]Geophysical Institute, University of Alaska Fairbanks, Fairbanks, 99775, USA
[2]Alaska Ecoscience, Fairbanks, 99709, USA

*Correspondence to*: W. L. Cable (wlcable@alaska.edu)

**Abstract.** Permafrost temperatures are increasing in Alaska due to climate change and in some cases permafrost is thawing and degrading. In areas where degradation has already occurred the effects can be dramatic, resulting in changing ecosystems, carbon release, and damage to infrastructure. Yet in many areas we lack baseline data, such as subsurface
temperatures, needed to assess future changes and potential risk areas. Besides climate, the physical properties of the vegetation cover and subsurface material have a major influence on the thermal state of permafrost. These properties are often directly related to the type of ecosystem overlaying permafrost. In this paper we demonstrate that classifying the landscape into general ecotypes is an effective way to scale up permafrost thermal data collected from field monitoring sites. Additionally, we find that within some ecotypes the absence of a moss layer is indicative of the absence of near surface
permafrost. As a proof of concept, we used the ground temperature data collected from the field sites to recode an ecotype landcover map into a map of mean annual ground temperature ranges at 1 m depth based on analysis and clustering of observed thermal regimes. The map should be useful for decision making with respect to land use and understanding how the landscape might change under future climate scenarios.

## 1 Introduction

Interest in permafrost as a potential source of the greenhouse gasses carbon dioxide and methane has increased, as we are beginning to understand the magnitude of the amount of carbon stored in these frozen soils (Koven et al., 2011; Schuur et al., 2015). However, measurements of the thermal state of permafrost, one of the main indicators of its stability, are sparse given the immense area underlain by permafrost (Romanovsky et al., 2010). It would be advantageous to use remote sensing and modeling to expand upon the direct measurements that are currently available. Satellite remote sensing of permafrost,
however, is complicated by the fact that currently there are no sensors that can penetrate the subsurface deep enough to make direct measurements of permafrost (National Research Council, 2014; Westermann et al., 2014). Instead, the presence or absence of permafrost and its thermal state must be inferred based on other parameters that can be remotely sensed such as land surface temperature (LST), topography, and vegetation through a combination of modeling and remote sensing.





Shur & Jorgenson (2007) have proposed a classification scheme for the formation and stability of permafrost based on the role of climate and ecosystem properties. This classification scheme points to the intimate relationship that exists between ecosystems and permafrost. The connection between permafrost and the atmosphere (in lowland areas) is not direct, rather its thermal state is influenced by vegetation, snow, surface water, soil properties, topography, and numerous interactions

between these components and by their interactions with permafrost (Jorgenson et al., 2010). It has long been known that vegetation plays an important role in the development and preservation of permafrost (Dingman and Koutz, 1974; Rieger et al., 1963; Stoeckeler, 1949; Viereck, 1970). Vegetation regulates the flux of energy into and out of the ground by controlling things such as the accumulation of organic layers and moss, and interception of solar radiation (Viereck, 1970). Viereck (1970) studied the formation of permafrost in a successional floodplain environment in central Alaska and found that

permafrost developed concurrently with the successional vegetation and began to appear as white spruce created conditions favorable for moss growth.

Mosses play an important role in permafrost formation and preservation due to their change in thermal conductivity depending on their moisture content and whether they are frozen or not. O'Donnell et al. (2009) found that dry live mosses had thermal conductivities between 0.02 and 0.04 W $m^{-1}$ $K^{-1}$, while water saturated mosses had thermal conductivities

approaching that of water, 0.56 W $m^{-1}$ $K^{-1}$ at 0°C (Lide, 2009), a more than tenfold increase. When frozen, the ice in these mosses would have a conductivity of 2.2 W $m^{-1}$ $K^{-1}$ at 0°C (Lide, 2009), a fourfold increase. This makes mosses more effective insulators during the summer than during the winter (Viereck, 1970). During the summer moss layers dry out, lowering their thermal conductivity and evaporation during this period also lowers the surface temperature. Then, during the fall as the air temperature cools, evaporation decreases, the moss layers become water saturated with late rainfall and early

snowfall events. As these saturated moss layers become frozen during the winter their thermal conductivity increases and this in turn increasing energy loss during the early winter before substantial snowfall accumulates (Viereck, 1970).

Snow is an excellent insulator, having thermal conductivity values between 0.08 W $m^{-1}$ $K^{-1}$ for new snow and 0.29 W $m^{-1}$ $K^{-1}$ for wind slab (Sturm et al., 2002). When sufficient accumulation of snow occurs mean annual ground temperatures can be increased by several degrees (Goodrich, 1982). However, total end of season snow depth is not the only thing that is

important. Early season snow accumulation is particularly important as this is when large amounts of latent heat are released as the active layer refreezes (Goodrich, 1982; Romanovsky and Osterkamp, 1995). The vegetation structure also influences snow accumulation through interception, primarily in spruce canopies (Viereck, 1970), and in the presence of wind through trapping of blowing snow (Sturm et al., 2001). Additionally, Sturm et al. (2001) found the deepest snow occurred in areas with the tallest, densest shrubs and that even small differences in the density of shrubs could have significant effects on snow

depth.

Aside from vegetation and snow, other properties are also important in controlling the way the overriding climate is translated to belowground temperatures including: hydrology, subsurface material, topography. These factors are often strongly associated with each other making it possible to identify distinct ecosystems and on a local scale these ecosystems can be classified into ecotypes (Jorgenson, 2000; Jorgenson et al., 2009). Ecotypes can be mapped from remotely sensed



data, such as Enhanced Thematic Mapper Plus and Thematic Mapper from Landsat, using the different spectral signatures created by vegetation composition and structure (Jorgenson et al., 2009). Thus, it seems reasonable that ecotypes could be used to infer properties of the underlying permafrost (or lack of permafrost).

The objectives of this paper are: (1) describe an established network of ground temperature monitoring sites in the Selawik area of north-west Alaska; (2) assess the climate gradient across the sites; (3) analyse the ground thermal regimes; and (4) develop a ground temperature map based on relationships between ground thermal regimes and ecotypes.

## 2 Research area and ecotype delineation

As an evaluation of ecotypes to infer permafrost characteristics, the Selawik National Wildlife Refuge (SNWR) in Western Alaska (Figure 1) was selected, as previously a high resolution ecotype map had been created for this area (Jorgenson et al., 2009). In addition, western Alaska in general, and the broad area centered on the SNWR and adjacent Bureau of Land Management (BLM) and National Park Service (NPS) lands in particular, were poorly represented in the network of permafrost temperature measurements developed in Alaska during the last 30 to 40 years by several scientific organizations. The permafrost temperature in this region has only been monitored in two relatively deep boreholes located near Nome and Kotzebue (60 and 29 m deep respectively). During the last several years, a network of shallow (2 to 6 m) boreholes has been established in the villages in this region as a part of the University of Alaska Fairbanks K-12 outreach program (Yoshikawa, 2013). However, this network is limited to locations near to local schools and does not represent the wide local variation in permafrost conditions in the region. Based on existing data, permafrost mean annual temperatures in Western Alaska vary generally between 0 and -4°C (most of existing data fall in the range between 0 and -2°C) and the permafrost spatial distribution changes from continuous in the north to no permafrost in the south (Figure 1). Existing observations show that as a result of recent warming local permafrost degradation has already started near the boundary of continuous and discontinuous permafrost, not only in Alaska but also in Siberia (Romanovsky et al., 2010). Present and future thawing of permafrost in these regions will have a dramatic effect on the ecosystems in this area because the permafrost generally has a high ice content, as a result of preservation of old, Late Pleistocene, ground ice in these relatively cold regions even during the warmer time intervals of the Holocene. The high vulnerability of the ecosystems to permafrost degradation in these transitional regions largely dictated our decision to begin establishment of a distributed permafrost observatory on the SNWR and adjacent BLM lands.

## 3 Methods

### 3.1 Establishment of Study Sites

Our study area, the SNWR, is located in Western Alaska (Figure 1). The SNWR covers 2.15 million acres and is named for the Selawik River that meanders through the middle of the refuge (U.S. Fish & Wildlife Service, 2003). In the fall of 2011,





sites for installation in summer 2012 were selected based on integrative analysis of the existing data on generalized ecotype classes (Figure 1), soil landscapes, and vegetation type distribution as documented in Jorgenson et al. (2009). Sites were selected to represent the most abundant ecotypes according to coverage dominance within the SNWR and to provide replication within the most abundant ecotypes (Table 1). In total, locations for 18 sites covering 11 of the 43 ecotypes and

5 two burned ecotypes were selected, representing 62.4% land area of the ecotypes within the SNWR. In addition to these 18 sites, three additional sites outside of the SNWR, previously installed in 2011, were included as they are within a similar climatic region as the SNWR. While we would have liked to include more measurement locations in order to cover more ecotypes and increase replication within ecotypes, this was not possible due to logistical and financial constraints. As there is no road access to SNWR, access during the summer is mostly by boat, airplane on floats, or helicopter. To be able to access

all areas of the refuge and not be limited to areas near waterways, we used a small helicopter (Robinson R44) to access most of our sites for installation of equipment and collection of data.

## 3.2 Measurement Design

Our measurement design consisted of a two-tiered site layout of core and distributed sites. The first tier of core sites, collected high temporal and vertical resolution temperature data. These sites comprised a CR1000 data logger (Campbell

Scientific, Logan, UT) that collected and saved data from the attached sensors measuring air temperature, snow depth, a high vertical resolution thermistor probe with 16 thermistors spaced exponentially to 1.5 m depth, and three deeper soil temperature sensors (2.0, 2.5 and 3.0 m in most cases). All temperature sensors were installed by drilling a small hole, approximately 2.5 cm in diameter, using a portable handheld hammer drill. The temperatures were measured every 5 minutes and hourly averages were stored on the data logger. The reported accuracy of the temperature sensors is 0.10 °C; however,

an ice bath calibration was carried out prior to sensor installation, improving the accuracy for temperatures near 0 °C to approximately 0.02 °C. The core sites were also equipped for remote communications using Iridium satellite transceivers or cellular modems and data was collected daily or weekly. Established in a transect from west to east, moving away from the ocean, and to cover a small elevational gradient (Figure 1, stars), these three sites allowed us to characterize any climatic dissimilarities that might be present within the study area.

To further characterize the climate within the area and to put our monitoring years in a historical context, we used daily summarized climate data from the Kotzebue Airport (OTZ) (Menne et al., 2012a, 2012b) located just to the west of the SNWR (Figure 1). Daily summarized air temperature and snow depth are available from this station beginning in 1946.

The second tier of distributed sites were deployed to capture the spatial variability in ground temperatures in the region (Figure 1). These sites consisted of a U-12 data logger (Onset, Cape Cod, Massachusetts) and four soil temperature sensors

located at 3, 50, 100, and 150 cm depth. At six sites it was not possible to drill to 150 cm due to rocks so the maximum sensor depth is 100 cm at four sites, 115 cm at one site, and 75 cm at one site. These data loggers record an instantaneous temperature every 4 hours. The reported accuracy of these temperature sensors is 0.25 °C; however, an ice bath calibration

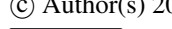



was performed prior to installation, improving the accuracy for temperatures near 0 °C to approximately 0.03 °C. Data from these sites has been collected manually once per year.

In 2013 during site visits to collect data, a small soil pit, approximately 30 by 30 cm, was excavated down to the top of permafrost or at least 75 cm at sites without near-surface permafrost. A general description of the soil profile was made for each site by dividing the soil layers into: living moss, litter, fibric organic material (slightly decomposed), humic organic material (moderate or highly decomposed), and mineral soil.

### 3.3 Data Analysis

Data analysis was conducted using MATLAB (R2013a, MathWorks Inc.). All raw data were first adjusted using a zero-offset that had been determined for each temperature sensor using an ice bath calibration in the lab before sensor installation. Erroneous values in the raw (hourly and 4 hour) data, due to sensor malfunctions, were detected visually and removed. Gaps in the raw data of up to 4 hours were filled using an average of the point's preceding and following the gap. Daily averages, minimums, and maximums were calculated from the raw data for days with at least 75% data coverage; gaps of two days or less in the time series of daily averages were filled using linear interpolation of the previous and following points. Gap filling of both raw and daily data was performed in only a few cases as most data was continuous and without erroneous values. Yearly averages, minimums, and maximums were calculated from the daily data only when 99% of the data was available to insure the data were not biased. A summary period from August 1st to July 31st of the following year was selected as this gave us a full year of data for analysis since the sites were installed in late July (summary periods 2011−2012 and 2012−2013). However, because in 2014 the sites were visited in late July, 10 July 2013 to 9 July 2014 was used as the 2013−2014 summary period in order to have a full year of data for this year.

Thaw depth was calculated from the daily mean subsurface temperatures at each site by first applying a 29 day moving average to smooth the data. The moving average acted to stabilize the near surface temperature (3 cm), but had little effect on the deeper depths as they were already filtered due to the natural damping of temperature variations that occurs with depth in the soil. Then, thaw depth was estimated daily at each site by fitting a piecewise cubic hermite polynomial to the daily temperatures with depth and evaluated at 0 °C for the depth of thaw penetration. This approach forced the temperature profile interpolation to pass through each measurement point, while preserving the shape of the temperature profile (Figure 2). It is important that the function fit to the data pass through each measurement point because at these points we know the temperature with the most certainty. Active layer thickness was defined as the maximum depth of the 0 °C isotherm for the entire warm (thawing) period of the year. To test the precision of this technique, the active layer thickness was computed at the three core sites using only 4 of the 16 temperature measurement depths. The resulting active layer thickness corresponded very well to what was estimated from the higher vertical resolution temperature measurements at these sites. An example of this comparison on the date of maximum thaw penetration (11 September 2013) at this site is shown in Figure 2, on the right the active layer thickness calculated using all 16 temperature sensors and on the left using only the 4 depths at the 2nd tier sites. The difference between two estimates in 2013 was 1 cm at the Kugurak Cabin (KC1) site and 3



cm at the Selawik Village (SV1). In 2012 the difference between the estimates was 1 cm at all three core sites. Furthermore, this validation shows that our choice of measurement depths, particularly with a measurement at 50 cm, is optimal for this area because the active layer thickness is often near 50 cm.

The timing of the active layer freeze-up was estimated to within a few days. The initiation of the freeze-back period was defined as the date when the daily mean temperature at the near surface (3 cm) dropped and remained below a threshold of -0.3°C for the rest of the season. This threshold was chosen because it has been shown in our previous investigations the temperature interval between 0 and -0.3°C represents the temperatures of major changes in the physical state of water during the freezing process in silty and organic soils (Romanovsky and Osterkamp, 2000). The end of the freeze-back period (time when the active layer was considered to be completely frozen) was defined as the date when all the temperature measurements had gone below this same threshold (e.g., 3 cm, 50 cm, 100 cm etc.).

To objectively evaluate the degree to which our sites were similar (or dissimilar) in terms of ground temperature dynamics, a cluster analysis was performed. A cluster analysis is a data based approach used to objectively classify data into groups where the within group dissimilarity is minimized and the between group similarly is maximized (Liao, 2005). This is in contrast to a more commonly used rule-based approach where groups are first defined arbitrarily for each measured quantity or quantities and then each measurement location is placed into a group (Fovell, 1997). One advantage of the data-based cluster analysis is that the classification rules do not have to be predefined, thus biases of the researcher are removed. For example, Fovell (1997) used this approach to delineate climate zones in the United States based on temperature and precipitation time-series data. Using the time-series of daily mean temperatures at 1 m from each of our sites and with missing data excluded, the pair-wise Euclidian distance between each site was computed. Then, the unweighted average Euclidian distance was used to create an agglomerative hierarchical cluster tree that could be visualized as a dendrogram. The total length of the U-shaped branches connecting two sites indicates the similarity of the datasets, where sites with small distances are most similar and sites with large distances are most dissimilar.

N-factors, which were originally developed by engineers as a way of estimating the freezing and thawing depth (Carlson, 1952; Lunardini, 1978), have also been applied in many studies of the natural environment (Jorgenson and Kreig, 1988; Kade et al., 2006; Karunaratne and Burn, 2004; Klene et al., 2001; Taylor, 1995). The n-factor, Eq. 1:

$$n = \frac{DD_s}{DD_a} \tag{1}$$

was calculated as the ratio of the degree-day sums of surface temperature ($DD_s$) to the degree-day sums of air temperature ($DD_a$). From our datasets, thawing and freezing n-factors were calculated using daily average air temperature and daily average surface temperature (3 cm depth) for each site and measurement period.

**3.4 Ground Temperature Map Development**

Based on the cluster analysis and the mean annual ground temperature (MAGT) at 1m depth from each ecotype, a map of MAGT was created using the ecotype delineations from Jorgenson et al. (2009). First, using ArcMap (version 10.1, ESRI)





each ecotype was recoded with the group number from the cluster analysis. For ecotypes where we did not have any measurements we used the vegetation and soil descriptions in Jorgenson et al. (2009) to group them with their most similar ecotype. Each cluster group was then assigned a range of expected MAGT at 1m depth: -4 to -1 °C, -2 to -1 °C, -1 to 0 °C, and greater than 0 °C. These ranges were chosen to accommodate the majority of MAGT ranges for each ecotype observed

during our measuring period. Additionally, a fifth, unknown, category was added for ecotypes that we were not comfortable classifying due to lack of information. Two versions of the MAGT map for the SNWR were created, one with all the ecotypes and one with only the ecotypes for which we made measurements.

## 4 Results

### 4.1 Climate Assessment

Measurements of the air temperature from our three core sites Selawik Village (SV1), Kugurak Cabin (KC1), and Selawik Thaw Slump (STS) (Figure 1) allow for comparison of how this parameter changes from the west to the east within the study area. This comparison shows that the seasonal changes in the air temperature are very similar for the SV1 and KC1 sites. The difference in mean monthly temperatures between these two sites does not exceed 2°C and is typically less than 1°C (Figure 3 & Figure 4, top). Comparison of the monthly means for our three sites to the monthly means for the Kotzebue airport

(OTZ) show good agreement during this measurement period (1 August 2012 to 31 July 2014).  Unfortunately our STS site stopped functioning in August 2013 due to wildlife damage so we do not have data for the 2013−2014 summary period. Mean annual air temperature calculated from OTZ and our three core sites show that on an annual basis temperatures are similar between sites (Table 2). The temperature at STS, however, is a little warmer compared to the other sites, which may be explained by slightly higher elevation of this site and presence of temperature inversions. The air temperature varies

substantially from year to year, however. The 2011−2012 measurement period was the coldest on average with temperatures close to the long-term (1981−2010) mean for OTZ with the exception of January 2012, which was considerably colder than the long-term mean. Air temperature during the 2012−2013 summary period shows that most months could be considered normal, with the exception of a cooler than normal November and December 2012 and slightly warmer June 2013 (Figure 3). During the 2013−2014 summary period, mean annual air temperatures were considerably warmer (Table 2), due in large

part to the considerably warmer months of October 2013 and January 2014, and slightly warmer April 2014 (Figure 4). In contrast to the air temperatures, available records from all three core sites show that the snow depths were anomalously low during the winter seasons of 2012−2013 and 2013−2014 (Figure 3 & Figure 4). These measurements agree well with the snow depth reported at OTZ and are well below the long-term (1981-2010) average. In 2012, the first substantial snowfall came very late in the season (mid-December) and by this time the active layer was already completely frozen at most sites.

In 2013, the first substantial snowfall also came later (early-November), but due to the warmer than average October the





active layer at most sites had just began to freeze. In contrast, during the 2011-1012 summary period the snow depth reported at OTZ was much higher than the long-term average (not shown).

**4.2 Ground Thermal Regime Analysis**

Ground temperature dynamics, as expected, were variable between sites and between measurement periods. For example, the

time-series of daily average ground temperature (3, 50, 100, and 150 cm depth) from two years (1 August 2012 to 31 July 2014) for three of our sites (KCF, KC1, and SV1) is presented in Figure 5. The time-series begins in August and surface temperatures (3 cm) are warm as the thaw depth approaches its maximum. As the surface temperature cools, the point at which it becomes negative signifies the beginning of the freeze-back period (Figure 5, red dashed line). The progression of the freezing front continues from the surface downward and the temperature at each depth remains constant, near 0 °C, until

the freezing front has passed. This effect of constant near-zero ground temperatures during the freezing period is termed the 'zero curtain'. When the freezing front passes a particular depth, the temperature at that depth decreases more rapidly, as almost all the liquid water at that depth has been converted to ice. Freeze-back is complete when all temperatures are below a threshold of -0.3°C (Figure 5, blue dashed line), indicating that the majority of liquid water has been frozen in the soil profile to the depth of our measurements. Finally, the point at which the 3-cm temperature becomes and stays positive signals the

beginning of the thawing period and the cycle begins again.

In this example of time-series data (Figure 5) distinct differences and similarities can be seen among sites and between years. For example, sites KC1 (Figure 5, B) and KCF (Figure 5, A) were only about 200 m apart, but were quite different in terms of their magnitude of temperature response and the timing of the active-layer refreezing. At site KC1 (Figure 5, B) freeze-back of the active layer was complete well before KCF (Figure 5, A). In contrast, sites KC1 (Figure 5, B) and SV1 (Figure 5,

C) were much more similar with respect to the magnitude of their temperature response and the date of active-layer refreezing, even though these sites were ~45 km apart. There were also differences between years within the same site, for example, in the winter of 2012–2013 the active layer at our three example sites was completely refrozen by early to mid-December; however, in the winter of 2013–2014 it didn't freeze back until mid-January or late-February. Thus, each time-series is like a unique fingerprint that is a result of the materials and processes occurring between the depth of the

temperature measurement and atmosphere above.

To determine the similarity and differences of ground temperature regimes among sites, independent of the ecotype classification, a hierarchical cluster analysis was performed. This analysis included all available daily averages of 1m ground temperature data from each of the 21 sites. The product was four distinct groups or clusters (Figure 6). Figure 8 and Figure 9 show temperature range, MAGT, and active layer thickness sorted according to the dendrogram and reveal that while groups

tend to have similar MAGT, the active layer thickness is somewhat more variable. With only one exception all sites of the same ecotype fell into the same cluster group, and we use this order for subsequent figures.

The freezing and thawing n-factors (Figure 7) are used to divide the effect of the vegetation and snow cover on the surface temperatures into freezing and thawing seasons. An n-factor near one indicates there is little difference between the air and



surface temperatures, while a thawing n-factor above one indicates a surface that is warmer than the air and a thawing n-factor below one indicates a surface that is colder than the air. The opposite is true for the freezing n-factor. In most natural systems n-factors are less than one due to the insulating effects and albedo of vegetation and snow (Taylor, 1995) and due to evaporation from the ground surface. The thawing n-factor gives us a relative sense of the amount of heat absorbed by the ground during the warm part of the year. While complicated to interpret, the freezing n-factor is related to the timing, thickness, and density of the snowpack. A thick snowpack would tend to keep the ground warmer producing a low freezing n-factor, while a thin or late snowpack would allow the surface temperature to more closely match the air temperature resulting in a freezing n-factor closer to 1. Figure 7 shows that the thawing n-factors for our sites generally fall between 0.8 and 1.0 and that between year differences for a given site are small. Thus, the insulative and cooling effects of the vegetation are more or less constant from year to years. The freezing n-factors show a much wider range of variation and a pronounced difference between our two measurement periods. The freezing n-factor in 2013−2014 for all sites was considerably lower than in 2012−2013, likely due to the late arrival of snow in winter 2012-2013. The freezing n-factors point to the importance of both the timing and depth of the snowpack in controlling the thermal regime.

The first group in the cluster analysis, with the coldest MAGT's, is composed mostly of the Upland Dwarf Birch-Tussock Shrub (TS) ecotype, which is abundant within the SNWR (28.4% areal coverage). The group also includes the Lowland Sedge Fen ecotype (SFL, 3.6% areal coverage) and Riverine Birch-Willow Low Shrub ecotype (BWR, 3.3% areal coverage), making the coverage of this grouping approximately 35% within the SNWR and the largest areal coverage of all the cluster groups. The vegetation within all of these ecotypes is primarily sedges and low shrubs, and with usually a thick moss layer (3−6 cm) that is underlain by a thick organic layer (fibric and humic) that often makes up most or all of the active layer (Figure 10). In 2012−2013, the MAGT at 1 m varied between -4.6 and -3.5 °C, while during 2013−2014 the MAGT was considerably warmer and varied between -2.8 and -0.8 °C (Table 3). The active layer was variable, but averaged 52 cm during both periods with the exception of the Tussock Post Burn site (S2-PB), which averaged 82 cm for the two years (Table 3). In 2012−2013, freeze-back of the active layer was complete by late November or early December, while in 2013−2014 freeze-back occurred in January or as late as March at one site (Table 3). The freezing n-factors (Figure 7) for these sites are the highest of all the cluster groups, indicating these sites have the best coupling to the atmosphere during the freezing season.

The second group identified by the cluster analysis was composed of three different ecotypes: Lowland Birch-Ericaceous Shrub (BEL, 7.3% areal coverage), Upland White Spruce-Ericaceous Forest (WSE, 4.8% areal coverage), and Upland Alder-Willow Tall Shrub (AWU, 4.4% areal coverage). Together, these ecotypes cover approximately 17% of the SNWR. The vegetation within this group was mostly low to medium shrubs with some sites having white spruce trees. The soil profile at these sites, like the first group, also tended to have a thick moss layer, but was underlain by somewhat thinner organic layers (fibric and humic). However, one site within the Lowland Birch-Ericaceous Shrub ecotype (site KCF) had only a thin (2 cm) leaf litter layer with no moss layer (Figure 10). The sites within this group have similar MAGT at 1 m, with a range of -3.2 to



-2.4 °C in 2012−2013 and -2.0 to -0.7 °C in 2013−2014 (Figure 8 & Figure 9), making them slightly warmer than the first group. The calculated active layer depths within this group were variable, averaging 66 cm in 2012−2013 and 46 cm in 2013−2014 (Table 3). The end of the freeze-back period was generally the same as group one, occurring by late November or early December in 2012−2013 and occurring later in 2013−2014 (Table 3). The freezing n-factors (Figure 7) for these

sites are similar but slightly lower than in the first group, indicating that sites in this group are also well coupled to the atmosphere during the freezing season.

The third group, with the warmest permafrost, was made up of only two ecotypes; the Lowland Alder-Willow Tall Shrub ecotype (AWL, 4.0% areal coverage) and the Upland Birch-Ericaceous Shrub ecotype (BEU, 3.2% areal coverage). Together these ecotypes occupy approximately 7% of the SNWR and formed the smallest group with near-surface permafrost.

Generally, the vegetation within these ecotypes had low to medium height shrubs and these sites had either a very thin or no moss layer underlain by organic layers similar in thickness to that of the second group (Figure 10). The MAGT at 1 m for these sites ranged from -1.9 to -1.1 °C in 2012−2013 and from -0.6 to -0.2 °C in 2013−2014 (Figure 8 and Figure 9). The active layer thickness and freeze back duration at these sites was variable (Table 3). The freezing n-factors (Figure 7) for these sites are lower than both of the first two groups and indicate that these sites are more decoupled from the atmosphere

during the freezing season, likely due to a thicker snowpack.

The fourth group identified in the cluster analysis included only the sites where we did not observe near-surface permafrost. This group is also the greatest distance from the other groups according to the cluster analysis (Figure 6). These sites belong to the Upland White Spruce-Willow Forest ecotype (WSW, 1.8% areal coverage), Upland Birch Forest (BFU, 0.6% areal coverage), and one site from the Upland Alder-Willow Tall Shrub (AWU, 4.4% areal coverage). Also included in this group

is a White Spruce site that had previously burned (WSB).  All these sites lack a moss layer on the surface and have a relatively thin organic layer (Figure 10). The freezing n-factors (Figure 7) at these sites are the lowest off all our sites and indicate they are the most decoupled from the atmosphere during the freezing season, likely due to a thicker and possibly earlier snowpack. Unfortunately, many of these sites had equipment malfunctions, making it difficult to calculate yearly summary statics (Figure 8, Figure 9 & Table 3). However, the ground temperature dynamics reflected in the available time-

series data for these sites indicates that permafrost is likely absent in the upper 1.5 m. Additionally, their clustering with sites known to lack near-surface permafrost lends support to this conclusion.

Based on our measurements freeze-back begins at approximately the same time across all sites, however, the duration often differs. During the 2012−2013 period the active layer began to freeze back in early October 2012 and freeze-up was complete at most sites by the beginning of December 2012 (Table 3). The very late and shallow snow-cover and related early

freeze-up of the active layer resulted in low winter, and thus annual, mean ground temperatures. During the 2013−2014 period freeze-back began much later (early-December 2013) and at some sites lasted until late-February or early-March 2014 (Table 3). Analysis of the mean annual ground temperatures at 1 m depth obtained from the measurement sites that were established in 2011 shows that the mean annual temperatures at this depth were lower in the 2012-2013 measurement period

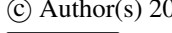



than in 2011-2012 by 1.5 to 1.8°C (Table 4). During the 2013−2014 measurement period MAGT at 1 m was the warmest of the three years (Table 4), which corresponds to the warmest mean annual air temperature. In general, the variation in MAGT at 1 m seen between years is as large as the variation among ecotypes.

### 4.3 Ground Temperature Map

As a proof of concept we used the range of MAGT at 1m depth measured across these different ecotypes (Table 3) and the clustering results to recode the ecotype map from Jorgenson et al. (2009) into a map of MAGT classes. Fortunately, our two main study years (2012−2013 and 2013−2014) included both a relatively cold (2012−2013) and warm (2013−2014) year allowing us to assess variability among years. We are confident these years bracket the longer-term mean ground temperature (and deeper permafrost temperature) because in 2012−2013 the slope of MAGT with depth was negative (Figure 8), indicating colder than average MAGT and mean annual air temperature (MAAT). While, in 2013−2014 the slope of

MAGT with depth was positive (Figure 9), indicating warmer than average MAGT and MAAT. While our measurements only covered 11 of the 43 ecotypes present in the SNWR, these ecotypes covered 62.4% of the land area in the SNWR. Two versions of the MAGT map for the SNWR were created, one with all the ecotypes (Figure 11) and one where the ecotypes we did not make any measurements in are masked out (Figure 12).

**5 Discussion**

A moss layer, which strongly affects soil temperatures, was not found in all of our ecotypes and this is possibly related to the presence of shrubs and trees in those ecotypes. When the density of deciduous trees and shrubs becomes sufficiently high, the annual leaf litter from these trees and shrubs can inhibit the growth of mosses (Viereck, 1970) by covering the ground and preventing the mosses from receiving light. However, this is not the case with coniferous species, which retain their

needles for longer periods of time.

We found that within the Upland Alder-Willow Shrub (AWU) ecotype and ecotypes containing White Spruce (WSW & WSE) there was a positive relationship between the presence of moss and the presence of near-surface permafrost. For example, within the White Spruce ecotypes the n-factors (Figure 7) can help us understand the difference between these sites. Within the Upland White Spruce-Willow Forest (WSW) ecotype our site (S1-WS), with no moss layer and no near-

surface permafrost, had low $n_f$ values; while the values of $n_t$ were similar to that of the Upland White Spruce Ericaceous (WSE) site (SS-WS), with a thick moss layer and permafrost. The WSE site, however, had much higher $n_f$ values, indicating that it was less insulated during the winter and was able to loose heat accumulated during the summer more readily. The same effect is likely occurring between our two AWU sites, but unfortunately we did not have sufficient surface temperature data from the AWU site without near-surface permafrost to calculate n-factors. The moss layer is important within other

ecotypes as well because it acts as an insulator during the summer keeping the thawing front from penetrating too deeply.




Tussocks in the Dwarf Birch-Tussock Shrub (TS) ecotype also have an important effect on the permafrost thermal regime. During the winter the tussocks stick up above the snow surface until enough snow has fallen to cover them completely. This creates holes in the snow cover, which would normally be a very good insulator, and allows heat to be removed from the ground surface by convecting air, cooling the ground. Additionally, these same tussocks have a shading effect during the summer, reducing the warming of the ground surface and permafrost. These factors work together to make tussock shrub ecotypes one of the coldest.

While there is some variability in n-factor values (Figure 7) within the cluster groups there are observations that can be made based on these values. We see that $n_t$ values generally range between 0.6 and 1.0 and there does not seem to be any relationship between ecotypes or cluster groups. The $n_f$ values however show a decreasing trend with increasing MAGT at 1 m. Cluster one, with the coldest MAGT, tends to have the highest $n_f$ values; while cluster four, with no near-surface permafrost, tends to have the lowest $n_f$ values. This indicates that the MAGT of an ecotype in this region depends more on how well it is able to release accumulated summer heat during the winter. There are exceptions to this generalization. Some sites in cluster one have low $n_f$ values; however, these sites also tend to have low $n_t$ values that would tend to offset this. There is also some variability between the two measurement periods, but almost all of this variability occurs during the freezing season. In fact, all the $n_f$ values are lower in 2013–2014 than they were in the previous year. This could be related to the late snowfall and early freeze-up of the active layer in 2012. With the active layer refrozen earlier in 2012 it would be a better conductor of heat to the surface for longer than during the following year, when the snow arrived earlier and the active layer refroze later.

The MAGT at 1m depth maps (Figure 11 and Figure 12) show that large areas of the SNWR, mainly the lowlands, are covered by ecotypes belonging to the coldest groups. These areas are probably more stable under a warming climate. However, areas along the rivers and streams and in the more upland areas tend to have warmer permafrost or lack permafrost entirely and are probably much more sensitive to any additional warming or disturbance. Evidence of areas with warmer permafrost can be found in the form of permafrost thaw features. One such feature, the Selawik Retrogressive Thaw Slump (RTS), is located along the Selawik River to the east and approximately 100 km upstream from Selawik (and near our site STS, Figure 11). The Selawik RTS formed in 2004 (USFWS, 2007) and the headwall has retreated at a rate of about 20 m/yr (Barnhart and Crosby, 2013). Closer inspection of the map in the area of the RTS indicates large areas classified as the warmest permafrost with smaller spots classified as no permafrost. Thus, maybe we can expect more of these features in this area as the climate continues to warm.

Closer inspection of the MAGT map around the Selawik River (e.g. inset in Figure 11) shows that areas immediately adjacent to the river belong either to the warmest permafrost group or lack near-surface permafrost. These areas, more recently modified by the meandering of the river, are in the early stages of vegetational succession and permafrost development. While areas that have not been modified by the river recently are classified into the colder permafrost groups. This agrees with what Viereck (1970) found in Interior Alaska, that newly fluvial modified surfaces did not have permafrost.





However, as the vegetation succession progresses, it begins to favor the formation of permafrost in later successional stages. It is uncertain though whether the climate will continue to favor the development of permafrost in these areas.

## 6 Conclusion

In this paper we have shown that ecotypes, which partition the variability in both vegetation and soil characteristics, are a reliable way to scale up observed ground thermal regimes from point to regional scale. This provides not only an opportunity for the scaling up of the ground thermal regime observed at field research sites but also for improved resolution of models of ground thermal regime. Accordingly, we recommend that future permafrost modeling efforts consider using an ecotype approach rather than more traditional grid-based approaches. Additional, future efforts to collect baseline ground temperature data should be focused on improving spatial coverage by establishing distributed sites in different ecotypes within a region.

Classification of the temperature time-series from our sites using a cluster analysis yielded four groups with distant properties. The first, coldest permafrost group, consisted mainly of ecotypes with sedges and low shrubs that tended to have thick moss and organic layers. The second, warmer permafrost group, contained mostly ecotypes with shorter shrubs or white spruce and also had a thick moss layer, but thinner organic layers than the first group. The third, warmest permafrost group, consisted mostly of ecotypes with tall shrubs and tended to have very thin or no moss layer and thinner organic layers. The fourth group, lacking permafrost within the top 1.5 m, had ecotypes with tall shrubs but lacked a moss layer and had thin organic layers. Thus, we find that an insulative moss layer is an important positive permafrost predictor. Warmer ground temperatures were associated with ecotypes with denser deciduous shrubs or trees, presumably because the shrubs and trees trap snow during the winter, which increases the snowpack, and generate more leaf litter, which reduces moss growth.

We used our results to generate a map of MAGT at 1m depth for the SNWR based on the ecotype landcover map produced by Jorgenson et al. (2009). This map shows that large areas in the lowlands of the SNWR are underlain by colder permafrost, while upland areas and areas adjacent to the rivers tend to be underlain by warmer or no permafrost at all.

Additionally, we collected a baseline of ground thermal data for the SNWR and surrounding areas which were previously underrepresented. We plan to continue collecting data from these sites as long as funding permits. The data used in this paper have been archived and are publicly accessible on the ACADIS Gateway (https://www.aoncadis.org/dataset/Permafrost_Western_AK_Selawik_NWR.html).

## Acknowledgements

The U.S. Fish and Wildlife Service and the Selawik National Wildlife Refuge, through Cooperative Ecosystem Studies Unit Agreement F11AC00613, supported this project. Additional support for this project was provided by NSF OPP grants ARC-





0856864 and -1304271. We thank the staff at the Selawik National Wildlife Refuge for help with logistics and lodging while conducting the fieldwork for this project. W. L. Cable thanks Bo Elberling and the Center for Permafrost (CENPERM), University of Copenhagen, Denmark for providing workspace to complete this manuscript.

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





**Figure 1** The map of the location of our research sites in the study area, the SNWR, north-west Alaska. The ecotype map (Jorgenson et al. 2009) is shown in the background where available.





**Table 1** The ecotype, ecotype areal coverage, and location of each site is shown in this table. Site codes in italics were installed in 2011 and are outside the SNWR. Site codes in bold are core sites.

| Site Code | Ecotype | Ecotype Code | Ecotype % Cover | Latitude | Longitude |
|---|---|---|---|---|---|
| *AKR* | Upland Dwarf Birch-Tussock Shrub | TS | 28.4 | 64.917500 | -160.728144 |
| *QZC* | Upland Dwarf Birch-Tussock Shrub | TS | 28.4 | 65.547459 | -161.403238 |
| S3-TM | Upland Dwarf Birch-Tussock Shrub | TS | 28.4 | 66.612523 | -158.655397 |
| S4-TM | Upland Dwarf Birch-Tussock Shrub | TS | 28.4 | 66.659274 | -160.121866 |
| **STS** | Upland Dwarf Birch-Tussock Shrub | TS | 28.4 | 66.501157 | -157.607440 |
| **SV1** | Upland Dwarf Birch-Tussock Shrub | TS | 28.4 | 66.605569 | -160.019213 |
| *UUG* | Upland Dwarf Birch-Tussock Shrub | TS | 28.4 | 65.055433 | -159.473368 |
| KCF | Lowland Birch-Ericaceous Low Shrub | BEL | 7.3 | 66.561726 | -159.000179 |
| S4-LS | Lowland Birch-Ericaceous Low Shrub | BEL | 7.3 | 66.655085 | -160.136155 |
| SS-WS | Upland White Spruce-Ericaceous Forest | WSE | 4.8 | 66.499779 | -157.604170 |
| S3-AWS | Upland Alder-Willow Tall Shrub | AWU | 4.4 | 66.611343 | -158.683565 |
| SS-AWS | Upland Alder-Willow Tall Shrub | AWU | 4.4 | 66.501420 | -157.609424 |
| S4-AWS | Lowland Alder-Willow Tall Shrub | AWL | 4.0 | 66.653454 | -160.148182 |
| S3-LSF | Lowland Sedge Fen | SFL | 3.6 | 66.584576 | -158.768248 |
| KCT | Riverine Birch-Willow Low Shrub | BWR | 3.3 | 66.562135 | -159.003357 |
| S3-BEW | Upland Birch-Ericaceous Low Shrub | BEU | 3.2 | 66.607057 | -158.679527 |
| S1-WS | Upland White Spruce-Willow Forest | WSW | 1.8 | 66.845685 | -160.017046 |
| **KC1** | Lowland Ericaceous Shrub Bog | ESB | 1.0 | 66.562380 | -159.004640 |
| S1-BF | Upland Birch Forest | BFU | 0.6 | 66.763641 | -160.092071 |
| S2-PB | Upland Burned Tussock Shrub | TSB | | 66.538220 | -158.362833 |
| S8-PB | Upland Burned White Spruce | WSB | | 66.891180 | -158.700893 |




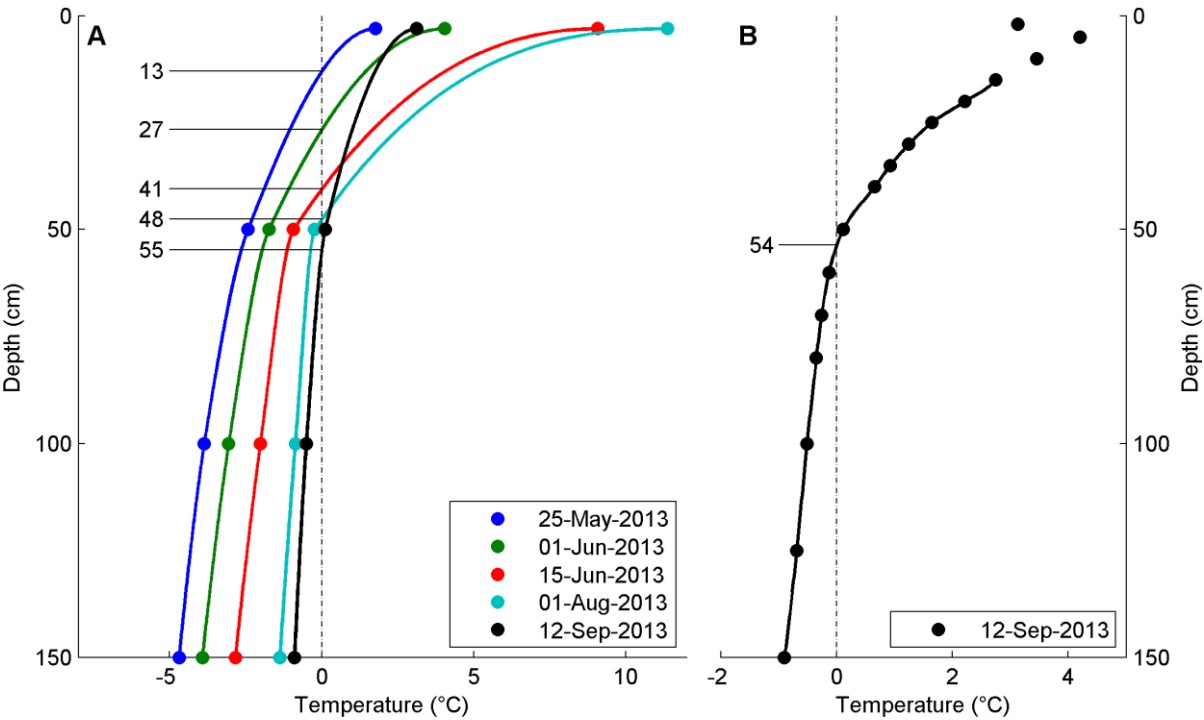

**Figure 2** Temperature depth profiles from site KC1. (A) Shows temperature depth profiles using only 4 depths for selected days with the estimated thaw depth to the left. (B) Shows the temperature depth profile with all 16 temperature measurements for the date near maximum thaw depth.

**Table 2** A summary of mean annual air temperature (MAAT) for our 3 study years from the Kotzebue Airport (OTZ), our Selawik Village site (SV1), Kugurak Cabin site (KC1), and Selawik Thaw Slump site (STS). The long-term average for OTZ is also shown.

| Year(s) | OTZ | SV1 | KC1 | STS |
|---|---|---|---|---|
| 1981−2010 | -5.09 | | | |
| 2011−2012 | -6.90 | | | |
| 2012−2013 | -5.30 | -5.74 | -6.05 | -4.69 |
| 2013−2014 | -2.41 | -3.14 | -3.14 | |



**Figure 3** Summary of air temperatures and snow depths for the period August 2012 to July 2013. The top panel shows the mean monthly air temperatures and standard deviations for our core sites and the Kotzebue (OTZ) airport; the blue boxes show the long-term (1981−2010) monthly means and standard deviations from the Kotzebue airport. The bottom panel shows the snow depths on the ground for our core sites and Kotzebue airport, with daily summary statistics for the same long-term period.



**Figure 4** Summary of air temperatures and snow depths for the period August 2013 to July 2014. The top panel shows the mean monthly air temperatures and standard deviations for our core sites and the Kotzebue (OTZ) airport; the blue boxes show the long-term (1981−2010) monthly means and standard deviations from the Kotzebue airport. The bottom panel shows the snow depths on the ground for our core sites and Kotzebue airport, with daily summary statistics for the same long-term period.







**Figure 5** Daily average temperatures at four depths from two years (Aug. 2012 to July 2014) is shown for three sites (A: KCF, B: KC1, & C: SV1). The start (red) and end (blue) of the freeze-back periods are identified.



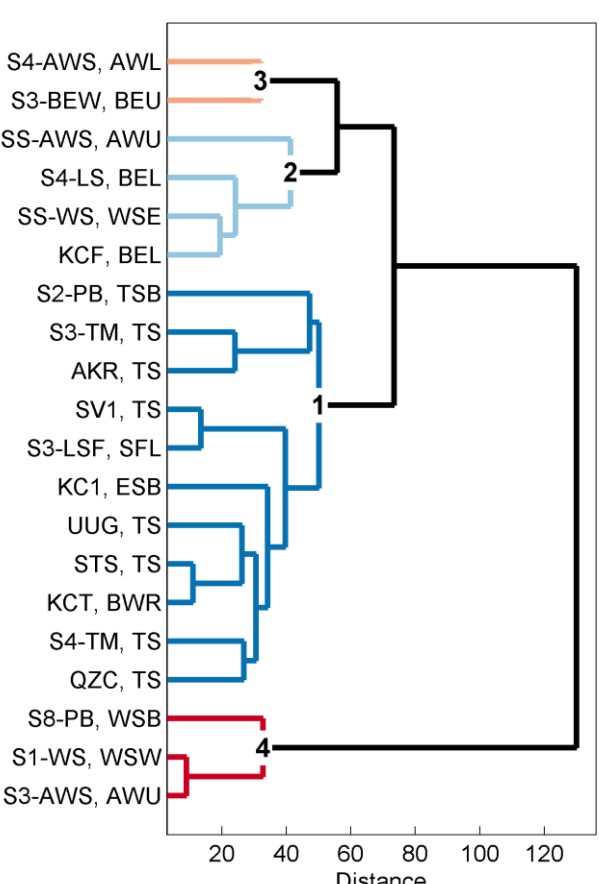

**Figure 6** The results of the cluster analysis are shown as a dendrogram.



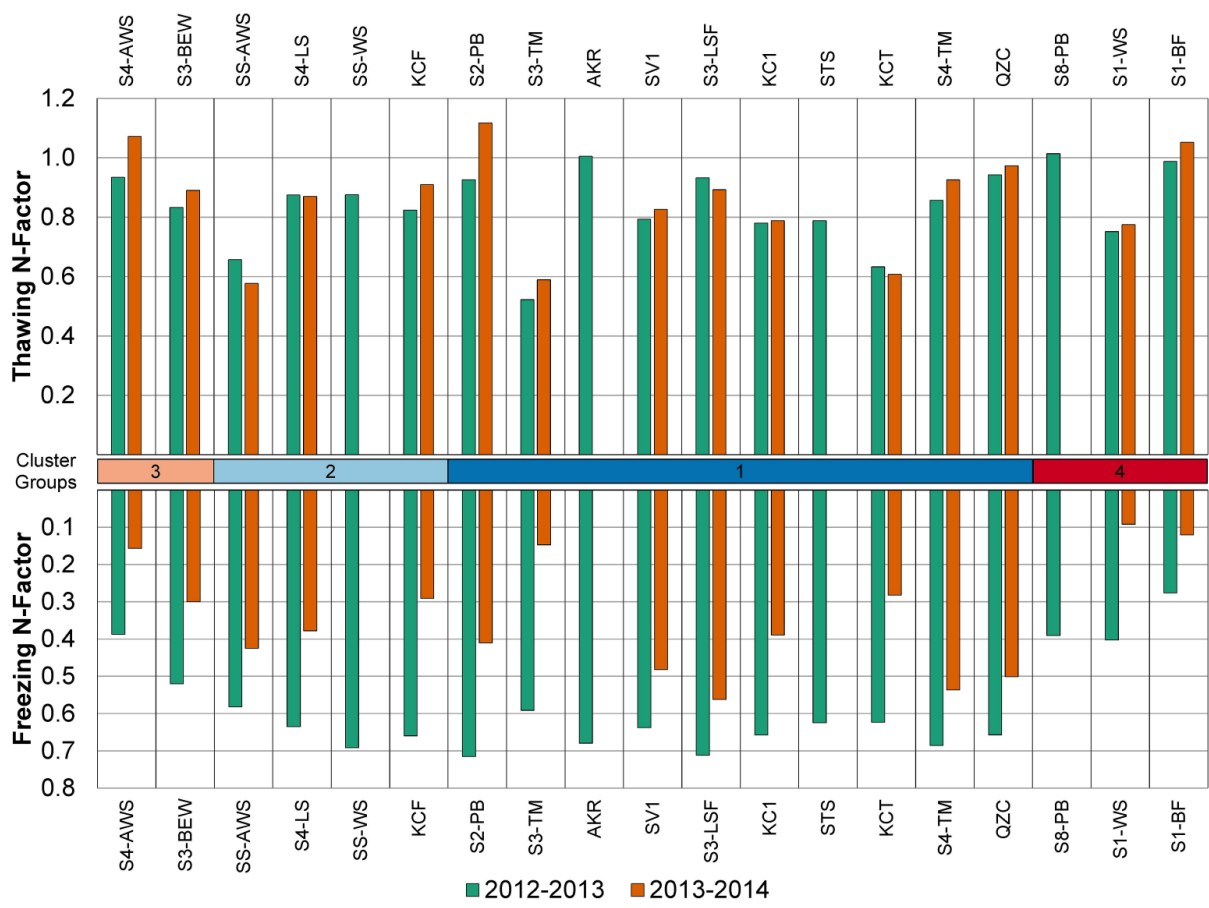

**Figure 7** Thawing n-Factors (top) and freezing n-Factors (bottom) for each site and measurement period.







**Figure 8** Annual summarized data for the period from 1 August 2012 to 31 July 2013. On the left is the annual mean (black squares) and range from daily averages (colored bars) for 3 depths from each site; in the center is the calculated active layer depth; and on the right the cluster analysis dendrogram for reference.







**Figure 9** Annual summarized data for the period from 1 August 2013 to 31 July 2014. On the left is the annual mean (black squares) and range from daily averages (colored bars) for 3 depths from each site; in the center is the calculated active layer depth; and on the right the cluster analysis dendrogram for reference.





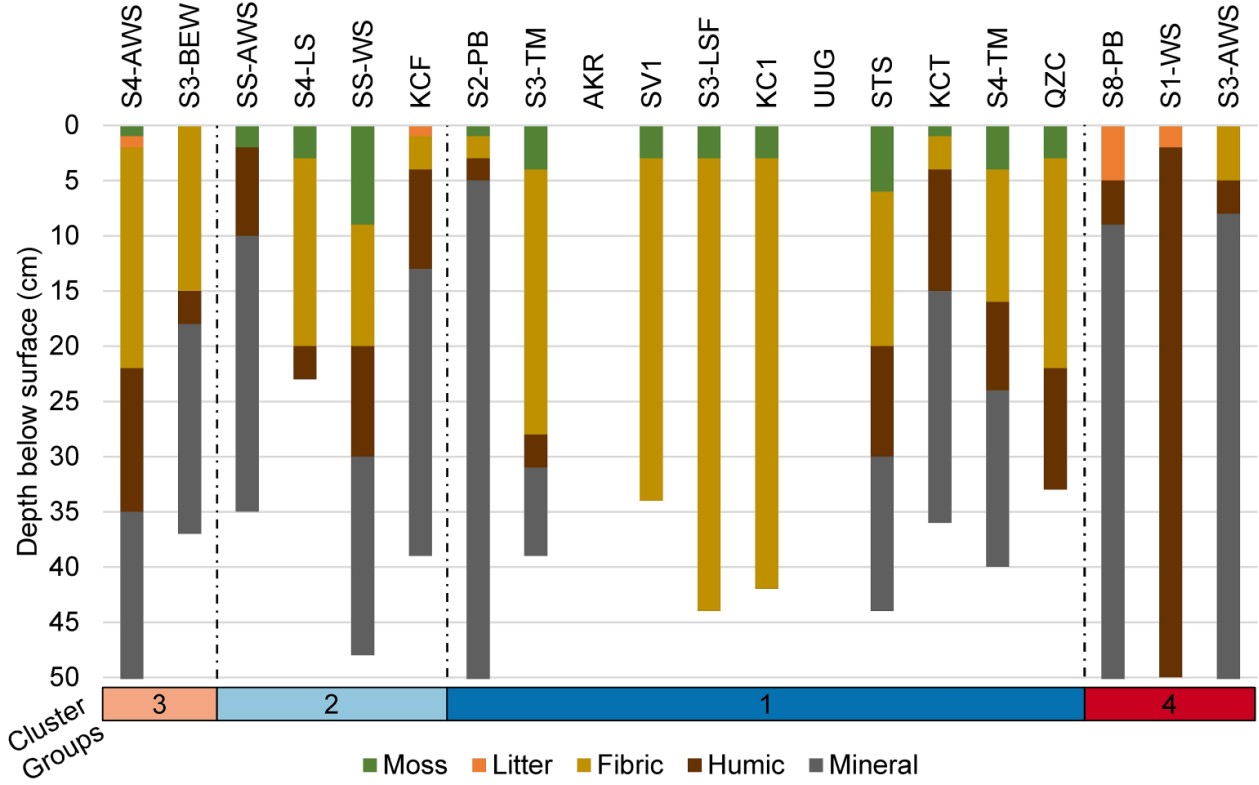

**Figure 10** The profiles of soil layers in the active layer at each site, organized according to the cluster analysis are shown.



**Table 3** A summary of the MAGT at 3 and 100 cm, the active layer depth, and the freeze-back date, for all study sites and for our two main measurement periods.

| Site Code | Ecotype Code | Cluster Group | 2012-2013 Measurement Period | | | | 2013-2014 Measurement Period | | | |
|---|---|---|---|---|---|---|---|---|---|---|
| | | | MAGT at 3cm (°C) | MAGT at 100cm (°C) | Active Layer (cm) | Freeze-Back Date | MAGT at 3cm (°C) | MAGT at 100cm (°C) | Active Layer (cm) | Freeze-Back Date |
| S4-AWS | AWL | 3 | -0.15 | -1.05 | 70 | 27-Dec | 3.00 | -0.20 | 69 | |
| S3-BEW | BEU | 3 | -1.66 | -1.92 | 48 | 11-Dec | 1.66 | -0.63 | 43 | 3-Mar |
| SS-AWS | AWU | 2 | -2.82 | -3.15 | 64 | 28-Nov | -0.38 | -1.96 | 41 | 22-Dec |
| S4-LS | BEL | 2 | -2.53 | -2.92 | 47 | 24-Nov | 0.97 | -1.27 | 47 | 16-Jan |
| SS-WS | WSE | 2 | -3.02 | -2.44 | 73 | 6-Dec | | | | |
| KCF | BEL | 2 | -2.92 | -2.64 | 80 | 20-Dec | 1.62 | -0.74 | 50 | 23-Feb |
| S2-PB | TSB | 1 | -3.05 | -4.38 | 84 | 30-Nov | 1.68 | -0.95 | 81 | 23-Feb |
| S3-TM | TS | 1 | -3.38 | -3.60 | 51 | 30-Nov | 1.29 | -0.81 | 44 | 14-Mar |
| AKR | TS | 1 | -2.46 | -3.52 | 51 | 5-Dec | | | | |
| SV1 | TS | 1 | -2.83 | -4.55 | 47 | 1-Dec | 0.20 | -2.57 | 55 | 10-Jan |
| S3-LSF | SFL | 1 | -3.00 | -4.56 | 68 | 22-Nov | -0.03 | -2.80 | 47 | 9-Jan |
| KC1 | ESB | 1 | -3.06 | -4.13 | 48 | 29-Nov | 0.60 | -1.76 | 53 | 21-Jan |
| UUG | TS | 1 | | | | | | | | |
| STS | TS | 1 | -2.74 | -3.92 | 48 | 1-Dec | | | | |
| KCT | BWR | 1 | -3.27 | -3.70 | 55 | 6-Dec | 0.57 | -1.37 | 47 | 10-Feb |
| S4-TM | TS | 1 | -3.03 | -4.29 | 54 | 26-Nov | 0.45 | -2.23 | 65 | 11-Jan |
| QZC | TS | 1 | -2.49 | -3.61 | 51 | 6-Dec | 0.62 | -1.92 | 51 | 29-Dec |
| S8-PB | WSB | 4 | | | | | | | | |
| S1-WS | WSW | 4 | -0.92 | 0.17 | | | 2.29 | -0.01 | | |
| S3-AWS | AWU | 4 | | 0.30 | | | | 0.02 | | |
| S1-BF | BFU | 4 | 1.02 | | | | 3.14 | | | |



**Table 4** The MAGT at 1 m depth for the three sites, installed in 2011, from which we have three years of data.

| Site | MAGT at 1 m depth | | |
|------|-----------|-----------|-----------|
|      | 2011–2012 | 2012–2013 | 2013–2014 |
| QZC  | -2.9      | -3.6      | -1.9      |
| KCT  | -2.0      | -3.7      | -1.4      |
| KCF  | -0.8      | -2.6      | -0.7      |

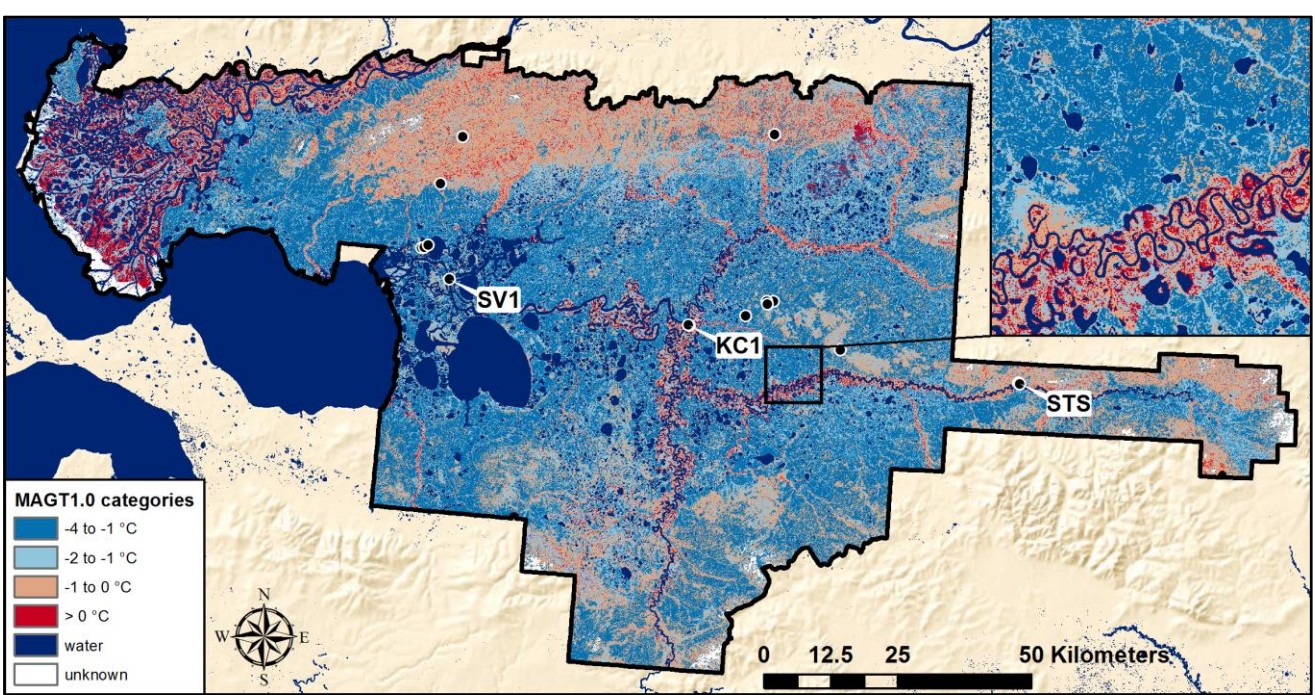

**Figure 11** Map of MAGT at 1m depth for the SNWR including estimates for unmeasured ecotypes.





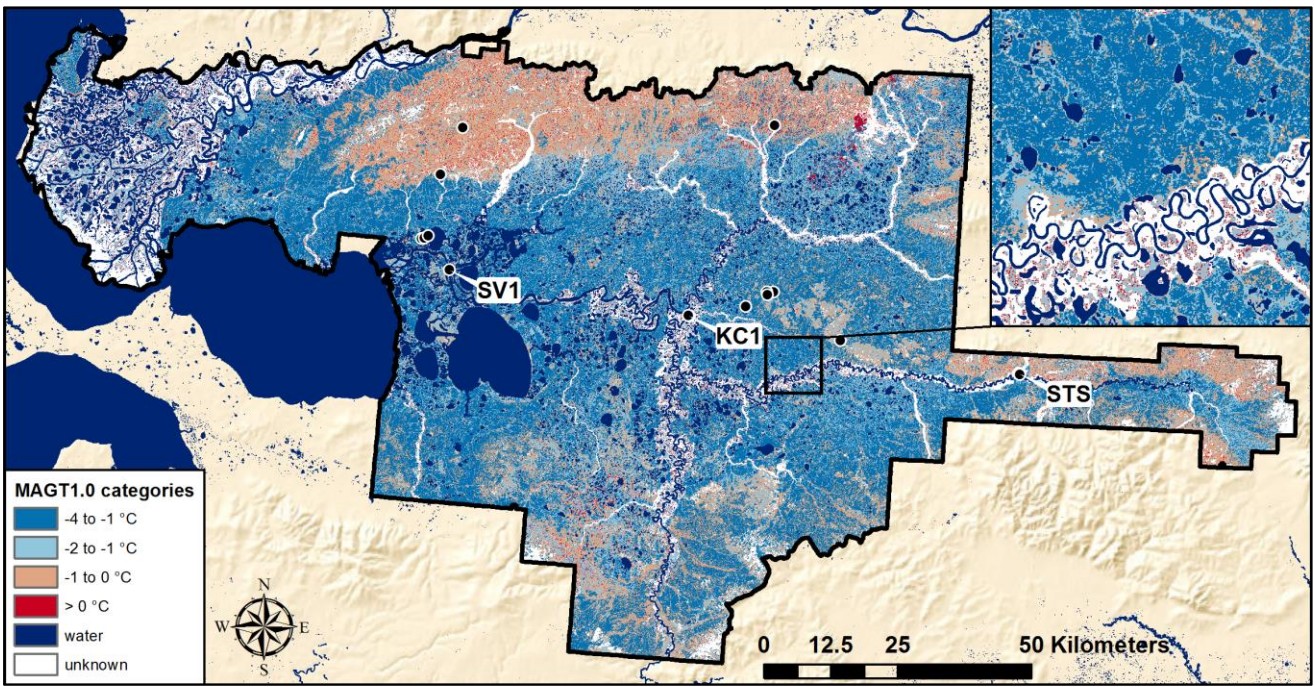

**Figure 12** Map of MAGT at 1m depth for the SNWR using only ecotypes for which we made measurements.