# Peer review of "Scaling-up Permafrost Thermal Measurements in Western Alaska using an Ecotype Approach"

_The Cryosphere, 2016_

## Referee Comment (RC1) · AW Balser (Referee) · 2 Apr 2016

This is a well-conceived and well-executed study, quite worthy of publication in The Cryosphere.

Permafrost, as a critical ecosystem component and factor in global climate impacts/feedbacks, must be better quantified spatially to enable improved estimates for: a) modes of permafrost degradation, b) impacts to landscapes and ecosystems, and c) carbon-based cumulative impacts to global climate. The authors rightly use well-developed ecotypes for this region as the basis for landscape-scale estimates of upper permafrost temperature and thermal properties based on rigorous field data. Ecotype currently comprises the best categorical scheme producible across remote landscapes which characterizes the most important surface and near-surface conditions influencing upper-permafrost properties and dynamics. In this case, study sites are primarily lowland locations, so ecotype alone should be sufficient to achieve the study goals presented here.

This manuscript represents an important early step toward broader development of both datasets and refined approaches for synoptic estimation of upper permafrost temperatures, and ultimately other key properties like ground ice and cryostructure distribution, at regional to global scales.

I have included a few suggestions for minor revisions/edits below. With one exception, none of them are essential to enabling publication of this work, but they're pretty easy changes if the authors agree with them, and may serve as improvements to the work.

The one exception is a point I strongly encourage the authors to reconsider (discussed under "Page13, Lines 7-8", and under "Page 12, Lines 19-24", below). The authors might possibly be well-justified in their recommendation to dispense with grid-based approaches for future work. However, if so, that justification isn't yet clear, and seems contrary to other successful approaches in the literature. If the authors prefer to retain this recommendation, the justification should really be better spelled out. Otherwise, I would have to respectfully disagree with their assertion in hopes that they remove it.

Page 1, Line 20: In the opening sentence of the introduction, the authors might include N2O along with CO2 and CH4. Nobody really talks much about it yet, since it's so poorly quantified in this context right now, but acknowledging the potential role of N2O might be a forward thinking inclusion here.

Page 11, Lines 8-11: This sounds like a bit of a strong statement given that there are really only two years of data. The authors might consider pulling back the language a bit to (very justifiably) claim they've captured some real inter-annual variability, without stating that it really brackets the long term variability.

Page 12, Lines 1-6: This is a really interesting point, with probable implications for

changing permafrost conditions following ecological shifts. If there happen to be any data, or other studies, addressing size/density of tussocks and how these impact thermal regime, it would be really interesting to mention them here in the discussion.

Page 12, Lines 19-24: The authors mention a few examples of effects from landscape position and aspect, without delving into it very deeply. Down the road, the best results from this sort of approach will likely include physiographic and geomorphologic variables along with ecotype in the analyses. I fully understand why the authors did not include them in this study, and I agree with their decision; including those variables here would have necessitated a number of field sites which would have been extremely prohibitive financially and logistically. Still, I think the end-game for this type of approach is to be able to cobble together enough congruent field data from enough projects and studies to enable such inclusion, and ultimately yield more precise results across landscapes and regions. It might do the readership a service to mention that explicitly here.

Page 13, Lines 7-8: While I agree that ecotype should represent the single most important variable in this sort of approach, and that ecotype alone is fully adequate in the context of estimates generated within this study, I don't recommend dispensing with a grid-based approach entirely for future work. There are a number of analytical techniques which can combine complementary categorical and continuous data to substantially improve results, and capture within-class variability very nicely through grid analyses. This can provide real advantages for testing ideas at multiple scales over using categorical units alone. Again, there's no reason for using a grid-based approach within this study, but I think if the authors want to stick with this recommendation for future work, it should probably include more justification as to why. There may be a good reason for this which I haven't considered, but if so, it would be important to describe it, given that grid-based approaches have provided a number of valuable contributions within other studies.

Figure 10: Extremely minor point - the color assignments for litter and for cluster group

3 are both orange. Given that there are a few colors not yet used in this figure, the authors might consider substituting one (purple, magenta or something). Would make it more quickly understood by the reader.

---

## Referee Comment (RC2) · A. Atchley (Referee) · 2 May 2016

The authors present a case for using ecotypes, which can be measured and spatially quantified using remote sensing techniques to assess the general state of permafrost. The idea and work presented here is of particular value to not just the permafrost community alone, but also biogeochemists and climate scientists that wish to understand the current state of the pan-Arctic permafrost and how changing ecological communities and permafrost co-evolve. The authors provide a well-articulated discussion that links ecotypes and plant community succession to the development of permafrost, both establishment and degradation. Here the authors use a cluster analysis to measure attributes, which then provides a nonsubjective approach to classifying the sites into categories. Though not to the extent of linking ecotype maps to permafrost as presented here, other studies in the Arctic have successfully used cluster analysis approaches

to link vegetation, elevation, organic layer thickness, surface hydrology in polygonal tundra permafrost environments, and therefore are worth mentioning. As correctly stated, if ecotypes and the direction of ecological succession are good diagnostic tools for permafrost conditions, then the combination of ecotype identification and remote sense can be used to evaluate subsurface permafrost conditions in sparsely monitored areas, such as the pan-Arctic. Therefore, I recommend this manuscript for publication in The Cryosphere Journal following minor revisions.

As stated above the authors motivate this work by providing a justifiable link between ecotype and permafrost establishment and degradation. However, the use of cluster analysis has been employed to link other landscape characteristics that can be measured using remote sensing to permafrost and carbon flux conditions. See introduction discussion in Wainwright et al (2015), which provides descriptions of other Arctic studies that employ zonation and cluster analysis to classify permafrost conditions to characteristics easily measured from remote sensing data (e.g Hinkel et al., 2003; Muster et al., 2012; Hubbard et al., 2013), some of which have noted that vegetation usually clusters well with other important thermal conditions. Plant communities or ecotypes are often related to landscape geomorphology, disturbance intervals, and many other environmental conditions. Furthermore, why this work is so compelling, at least to me, is that ecotypes themselves as well as the plant community sessional stage can be a product of these combined conditions and therefore may well function as a system condition integrator. In my opinion it would be beneficial to the cyrosphere community if the authors also included a discussion about how the ecotype classification differs or adds to the work that links other landscape characteristics to permafrost conditions.

Hinkel, Kenneth M., et al. "Spatial extent, age, and carbon stocks in drained thaw lake basins on the Barrow Peninsula, Alaska." Arctic, Antarctic, and Alpine Research 35.3 (2003): 291-300.

Hubbard, Susan S., et al. "Quantifying and relating land-surface and subsurface variability in permafrost environments using LiDAR and surface geophysical datasets." Hydrogeology Journal 21.1 (2013): 149-169.

Muster, Sina, et al. "Subpixel heterogeneity of ice-wedge polygonal tundra: a multi-scale analysis of land cover and evapotranspiration in the Lena River Delta, Siberia." Tellus B 64 (2012).

Wainwright, Haruko M., et al. "Identifying multiscale zonation and assessing the relative importance of polygon geomorphology on carbon fluxes in an Arctic tundra ecosystem." Journal of Geophysical Research: Biogeosciences 120.4 (2015): 788-808.

My second somewhat major suggestion is to re-organize the result section of the paper. To me the most important results of the manuscript start on page 9 Line 14 and go to the end of the results section including section 4.3, which is buried in the middle of the results section. Following the top down approach of technical writing, where the main result and conclusion should be presented first, I would move these results to the top of the section before section 4.1. Section 4.1 and the first half of Section 4.2 'Ground Thermal Regime Analysis seem to be out of place and I would consider them only supporting results to the main message, which is how ecotypes and permafrost conditions are linked, which then produces the ground temperature map. This of course is my preference in technical writing (and reading for that matter), which I hope will help improve an already good paper and increase its impact.

The following are minor suggestions that I hope will improve the quality of the manuscript.

1) Page2 Line16: Add 'compared to water' to the end of the sentence '...a fourfold increase.'

2) Page2 Line23-24: Change '...annual ground temperatures can be increased by several...' to '...annual ground temperatures can increase by several...'

3) Page2 Line24-25: Sentence is not needed, "However, total end of season snow depth is not the only thing that is important."

4) Page2 Line 30: May help to specifically point out to the reader that increased snow depth, which insulates the ground in winter will lead to warmer permafrost temps. Likewise on Page2 Line 21, may help to specifically point out that moss will cool the subsurface leading to colder permafrost. I believe that is the point that these paragraphs are making, and therefore should be stated clearly.

5) Page3 Line 21-24: "Present and future thawing of permafrost in these regions will have a dramatic effect on the ecosystems in this area because the permafrost generally has a high ice content, as a result of preservation of old, Late Pleistocene, ground ice in these relatively cold regions even during the warmer time intervals of the Holocene." How does the preservation of the old cold regions affect the ecosystems? This sentence seems to have 2 separate messages that are may be unrelated.

6) Page4 Line 8-11: Are two sentences describing how the plots were accessed necessary? Perhaps rephrase to only one sentence, "Due to the remote nature and inaccessibility of the sites by road, a small helicopter (Robinson R44) was used to access areas in the refuge beyond the reach of waterways." By the way, the helicopter bit is pretty cool!

7) Page5 Line21-23: At this point it is not clear that the near surface temperature (3cm) is an important part of the analysis, and the 29 day moving average seems unnecessary. Latter in the paper it becomes clear that you do use it. Perhaps it would help if before this point some mention of why near surface temps are important and that they fluctuate a lot was added.

8) Page5 Line 26: The phrase, "...function fit to data pass through each measurement point..." is awkward, try to rephrase.

9) Page5 Line 31: '...at this site is shown...' Replace 'this' with KC1. Also I noticed throughout the manuscript that 'this' is used a lot, when it would help to be more specific and clearer to say what 'this' is.

10) Page6 Line 17: Again 'this' in "Fovell (1997) used this approach" is vague. Did Fovell use the cluster or rule-based approach?

11) Page6 Line 20: May be helpful to reference figure 6 here for an example of a dendrogram.

12) Page12 Line1-6: The tussock discussion is interesting in that it details how plants and ecosystems can govern environmental conditions. My question is, wouldn't the thermal conductivity of the tussock have to be high or relatively higher than snow to able to conduct energy from the subsurface to the atmosphere in order for the winter cooling affect to happen. While not within the scope of this paper, modeling schemes maybe able to define what thermal conductivities of tussocks are necessary to have a cooling effect, or what densities of tussocks sticking up above the snow are necessary.

13) Page12 Line29-33: The discussion of the interaction between the river disturbance and plant community succession is an important result/discussion point of the paper as it provides another example of 1) the interaction between geomorphology and ecology, and 2) how plant community succession determines the physical environment (i.e subsurface temperature). I would suggested that this point be highlighted more as it could provide further evidence as to 1) why ecotype classification can be used to map permafrost conditions and 2) that understanding the interaction of disturbance and the direction of plant community succession will help inform permafrost evolution.

14) Page13 Line 8: What do you mean by grid-based approaches? Finite difference and finite-volume or spatially distributed GCM's models come to mind. CLM and many spatially distributed models have plant functional type representation and ways of simulating ecotypes and the effects of those types on permafrost. Here I agree that models should link ecological types to the physical environment, but what is to limit grid based models from doing this?

15) Page12 Line 25: Is it appropriate to bring up funding here? Financial constraints at some point limit most studies as has already been acknowledge on page 4 line 8,

but is this publication the appropriate place to discuss the lack of money in sciences? I know Robinson R44 helicopters are expensive, despite being supper cool. However, it may be better to discuss the benefits of continued and additional data gathering, which would then provide motivation for continued funding. How might more measurements build confidence in the ecotype approach and reduce uncertainty in permafrost assessment.

---

## Author Comment (AC1) · 18 Jun 2016

**Response to Review #1 by A. W. Balser on, "Scaling-up Permafrost Thermal Measurements in Western Alaska using an Ecotype Approach"**

William L. Cable[1], Vladimir E. Romanovsky[1], M. Torre Jorgenson[2]

[1]Geophysical Institute, University of Alaska Fairbanks, Fairbanks, 99775, USA
[2]Alaska Ecoscience, Fairbanks, 99709, USA

*Correspondence to*: W. L. Cable (wlcable@alaska.edu)

We would like to thank A. W. Balser for his helpful review of our manuscript. We agree with his comments and have revised the manuscript accordingly. Below, A. W. Balser's comments are given in italics and our response as regular text in blue.

*This is a well-conceived and well-executed study, quite worthy of publication in The Cryosphere.*

*Permafrost, as a critical ecosystem component and factor in global climate impacts/feedbacks, must be better quantified spatially to enable improved estimates for: a) modes of permafrost degradation, b) impacts to landscapes and ecosystems, and c) carbon-based cumulative impacts to global climate. The authors rightly use well developed ecotypes for this region as the basis for landscape-scale estimates of upper permafrost temperature and thermal properties based on rigorous field data. Ecotype currently comprises the best categorical scheme producible across remote landscapes which characterizes the most important surface and near-surface conditions influencing upper-permafrost properties and dynamics. In this case, study sites are primarily lowland locations, so ecotype alone should be sufficient to achieve the study goals presented here.*

*This manuscript represents an important early step toward broader development of both datasets and refined approaches for synoptic estimation of upper permafrost temperatures, and ultimately other key properties like ground ice and cryostructure distribution, at regional to global scales.*

*I have included a few suggestions for minor revisions/edits below. With one exception, none of them are essential to enabling publication of this work, but they're pretty easy changes if the authors agree with them, and may serve as improvements to the work.*

*The one exception is a point I strongly encourage the authors to reconsider (discussed under "Page13, Lines 7-8", and under "Page 12, Lines 19-24", below). The authors might possibly be well-justified in their recommendation to dispense with grid-based approaches for future work. However, if so, that justification isn't yet clear, and seems contrary to other successful approaches in the literature. If the authors prefer to retain this recommendation, the justification should really be better spelled out. Otherwise, I would have to respectfully disagree with their assertion in hopes that they remove it.*

*Page 1, Line 20: In the opening sentence of the introduction, the authors might include N2O along with CO2 and CH4. Nobody really talks much about it yet, since it's so poorly quantified in this context right now, but acknowledging the potential role of N2O might be a forward thinking inclusion here.*

We agree and have included $N_2O$ in the text.

"Interest in permafrost as a potential source of the greenhouse gasses $CO_2$, $CH_4$, and $N_2O$ has increased…"

*Page 11, Lines 8-11: This sounds like a bit of a strong statement given that there are really only two years of data. The authors might consider pulling back the language a bit to (very justifiably) claim they've captured some real inter-annual variability, without stating that it really brackets the long term variability.*

The language in this statement has been pulled back a little however, we feel the support for this statement is quite conclusive given the permafrost temperature at depth represents a long-term average.

"We think these years likely bracket the longer-term mean ground temperature (and deeper permafrost temperature) because in 2012–2013 the slope of MAGT with depth was negative (Figure 8), indicating colder than average MAGT and mean annual air temperature (MAAT)."

*Page 12, Lines 1-6: This is a really interesting point, with probable implications for C2 changing permafrost conditions following ecological shifts. If there happen to be any data, or other studies, addressing size/density of tussocks and how these impact thermal regime, it would be really interesting to mention them here in the discussion.*

We also find this to be a very interesting point and have observed this many times while visiting field sites early in the winter. Unfortunately though, we are unaware of any data or studies addressing the size/density of tussocks and how this would impact the thermal regime.

*Page 12, Lines 19-24: The authors mention a few examples of effects from landscape position and aspect, without delving into it very deeply. Down the road, the best results from this sort of approach will likely include physiographic and geomorphologic variables along with ecotype in the analyses. I fully understand why the authors did not include them in this study, and I agree with their decision; including those variables here would have necessitated a number of field sites which would have been extremely prohibitive financially and logistically. Still, I think the end-game for this type of approach is to be able to cobble together enough congruent field data from enough projects and studies to enable such inclusion, and ultimately yield more precise results across landscapes and regions. It might do the readership a service to mention that explicitly here.*

We fully agree that in some areas ecotypes might not be relevant or completely explain the variation in permafrost thermal regime. We didn't feel that this (page 12, lines 19-24) was the

right place to address this so a sentence has been added to the conclusion (page 13, lines 9-11) that addresses this.

"However, in some areas (e.g. mountainous terrain or barren landscapes), variables other than ecotypes (e.g. slope, aspect, or microtopography) may become more important, in which case they could be used in addition to, or instead of ecotypes."

*Page 13, Lines 7-8: While I agree that ecotype should represent the single most important variable in this sort of approach, and that ecotype alone is fully adequate in the context of estimates generated within this study, I don't recommend dispensing with a grid-based approach entirely for future work. There are a number of analytical techniques which can combine complementary categorical and continuous data to substantially improve results, and capture within-class variability very nicely through grid analyses. This can provide real advantages for testing ideas at multiple scales over using categorical units alone. Again, there's no reason for using a grid-based approach within this study, but I think if the authors want to stick with this recommendation for future work, it should probably include more justification as to why. There may be a good reason for this which I haven't considered, but if so, it would be important to describe it, given that grid-based approaches have provided a number of valuable contributions within other studies.*

We have removed the implication that our "ecotype approach" should be used instead of a grid-based approach and suggested instead that the ecotype approach offers an improvement in spatial resolution without increased computational demand.

"Accordingly, we recommend that future permafrost modeling efforts consider using an ecotype approach as it offers increased spatial resolution without increased computational demand (i.e. a model only needs to be run once for each ecotype)."

*Figure 10: Extremely minor point - the color assignments for litter and for cluster group C3 3 are both orange. Given that there are a few colors not yet used in this figure, the authors might consider substituting one (purple, magenta or something). Would make it more quickly understood by the reader.*

Thank you for the suggestion, the color of litter in Figure 10 (below) has been changed so it is easier to distinguish from cluster group 3.

---

## Author Comment (AC2) · 18 Jun 2016

**Response to Review #2 on, "Scaling-up Permafrost Thermal Measurements in Western Alaska using an Ecotype Approach"**

William L. Cable[1], Vladimir E. Romanovsky[1], M. Torre Jorgenson[2]

[1]Geophysical Institute, University of Alaska Fairbanks, Fairbanks, 99775, USA
[2]Alaska Ecoscience, Fairbanks, 99709, USA

*Correspondence to*: W. L. Cable (wlcable@alaska.edu)

We would like to thank A. Atchley for his helpful review of our manuscript. We agree with most of his comments and have revised the manuscript accordingly. Below, A. Atchley's comments are given in italics and our response as regular text in blue.

*The authors present a case for using ecotypes, which can be measured and spatially quantified using remote sensing techniques to assess the general state of permafrost. The idea and work presented here is of particular value to not just the permafrost community alone, but also biogeochemists and climate scientists that wish to understand the current state of the pan-Arctic permafrost and how changing ecological communities and permafrost co-evolve. The authors provide a well-articulated discussion that links ecotypes and plant community succession to the development of permafrost, both establishment and degradation. Here the authors use a cluster analysis to measure attributes, which then provides a nonsubjective approach to classifying the sites into categories. Though not to the extent of linking ecotype maps to permafrost as presented here, other studies in the Arctic have successfully used cluster analysis approaches to link vegetation, elevation, organic layer thickness, surface hydrology in polygonal tundra permafrost environments, and therefore are worth mentioning. As correctly stated, if ecotypes and the direction of ecological succession are good diagnostic tools for permafrost conditions, then the combination of ecotype identification and remote sense can be used to evaluate subsurface permafrost conditions in sparsely monitored areas, such as the pan-Arctic. Therefore, I recommend this manuscript for publication in The Cryosphere Journal following minor revisions.*

*As stated above the authors motivate this work by providing a justifiable link between ecotype and permafrost establishment and degradation. However, the use of cluster analysis has been employed to link other landscape characteristics that can be measured using remote sensing to permafrost and carbon flux conditions. See introduction discussion in Wainwright et al (2015), which provides descriptions of other Arctic studies that employ zonation and cluster analysis to classify permafrost conditions to characteristics easily measured from remote sensing data (e.g Hinkel et al., 2003; Muster et al., 2012; Hubbard et al., 2013), some of which have noted that vegetation usually clusters well with other important thermal conditions. Plant communities or ecotypes are often related to landscape geomorphology, disturbance intervals, and many other environmental conditions. Furthermore, why this work is so compelling, at least to me, is that*

*ecotypes themselves as well as the plant community sessional stage can be a product of these combined conditions and therefore may well function as a system condition integrator. In my opinion it would be beneficial to the cyrosphere community if the authors also included a discussion about how the ecotype classification differs or adds to the work that links other landscape characteristics to permafrost conditions.*

*Hinkel, Kenneth M., et al. "Spatial extent, age, and carbon stocks in drained thaw lake basins on the Barrow Peninsula, Alaska." Arctic, Antarctic, and Alpine Research 35.3 (2003): 291-300.*

*Hubbard, Susan S., et al. "Quantifying and relating land-surface and subsurface variability in permafrost environments using LiDAR and surface geophysical datasets." Hydrogeology Journal 21.1 (2013): 149-169.*

*Muster, Sina, et al. "Subpixel heterogeneity of ice-wedge polygonal tundra: a multiscale analysis of land cover and evapotranspiration in the Lena River Delta, Siberia." Tellus B 64 (2012).*

*Wainwright, Haruko M., et al. "Identifying multiscale zonation and assessing the relative importance of polygon geomorphology on carbon fluxes in an Arctic tundra ecosystem." Journal of Geophysical Research: Biogeosciences 120.4 (2015): 788-808.*

We agree, a discussion of other studies that have used cluster analysis might be useful, thank you for pointing out a few examples. We have added a paragraph to the beginning of the Discussion to discuss studies that have previously used clustering analysis as well as our use of clustering analysis.

"We used a clustering approach to classify each site based on the daily time-series at 1 m depth. A clustering or zonation approach has been used before in Arctic studies (e.g. Hinkel et al., 2003; Hubbard et al., 2013; Muster et al., 2012; Wainwright et al., 2015), but never before using a ground temperature time-series, as was done in this study. A similar approach was taken by Hubbard et al. (2013) and Wainwright et al. (2015) using geophysical and remotely sensed data as input to the cluster analysis. Other studies (e.g. Hinkel et al., 2003; Zona et al., 2011) however, have only used spatial, remotely sensed data to classify the spatial heterogeneity (vegetation, microtopography, etc.) into landscape classes and then tested for correlations among measured parameters within these classes. While we also used a landscape classification, ecotypes, our cluster analysis was based solely on the ground temperature dynamics data from each site, independent of the sites ecotype. Using a cluster analysis in this way is beneficial because it removes any judgement from the researcher as to how the data should be grouped. This approach reinforced our use of ecotypes to scale up ground thermal measurements as each group included sites of the same and similar ecotypes."

*My second somewhat major suggestion is to re-organize the result section of the paper. To me the most important results of the manuscript start on page 9 Line 14 and go to the end of the results section including section 4.3, which is buried in the middle of the results section. Following the top down approach of technical writing, where the main result and conclusion should be presented first, I would move these results to the top of the section before section 4.1.*

*Section 4.1 and the first half of Section 4.2 'Ground Thermal Regime Analysis seem to be out of place and I would consider them only supporting results to the main message, which is how ecotypes and permafrost conditions are linked, which then produces the ground temperature map. This of course is my preference in technical writing (and reading for that matter), which I hope will help improve an already good paper and increase its impact.*

On the reorganization of the results section we respectfully disagree. The results are organized so that supporting results, needed to understand the main results, are presented first. For example, the results of the climate analysis are presented first because they are important in interpreting the results from the ground thermal regime.

*The following are minor suggestions that I hope will improve the quality of the manuscript.*

*1) Page2 Line16: Add 'compared to water' to the end of the sentence '. . .a fourfold increase.'*

Agreed, added.

*2) Page2 Line23-24: Change '. . .annual ground temperatures can be increased by several. . .' to '. . .annual ground temperatures can increase by several. . .'*

Agreed, changed.

*3) Page2 Line24-25: Sentence is not needed, "However, total end of season snow depth is not the only thing that is important."*

Agreed, removed.

*4) Page2 Line 30: May help to specifically point out to the reader that increased snow depth, which insulates the ground in winter will lead to warmer permafrost temps. Likewise on Page2 Line 21, may help to specifically point out that moss will cool the subsurface leading to colder permafrost. I believe that is the point that these paragraphs are making, and therefore should be stated clearly.*

This is a good point, while these two paragraphs point out the importance of mosses and snow it was not explicitly stated. A sentence has been added to the end of each paragraph to state this clearly.

"Thus, addition of or increasing the thickness of moss layers generally leads to lower permafrost temperatures."

"Therefore, increasing the depth and duration of snow cover generally leads to increased ground temperatures."

*5) Page3 Line 21-24: "Present and future thawing of permafrost in these regions will have a dramatic effect on the ecosystems in this area because the permafrost generally has a high ice content, as a result of preservation of old, Late Pleistocene, ground ice in these relatively cold regions even during the warmer time intervals of the Holocene." How does the preservation of the old cold regions affect the ecosystems? This sentence seems to have 2 separate messages that are may be unrelated.*

We believe this sentence is clearly stated, with the main message being the high ground ice content in this region, supported by why there is a high ground ice content.

*6) Page4 Line 8-11: Are two sentences describing how the plots were accessed necessary? Perhaps rephrase to only one sentence, "Due to the remote nature and inaccessibility of the sites by road, a small helicopter (Robinson R44) was used to access areas in the refuge beyond the reach of waterways." By the way, the helicopter bit is pretty cool!*

Agreed, this was shortened as suggested.

"Due to the remote nature and inaccessibility of the sites by road, a small helicopter (Robinson R44) was used to access areas in the refuge beyond the reach of waterways."

*7) Page5 Line21-23: At this point it is not clear that the near surface temperature (3cm) is an important part of the analysis, and the 29 day moving average seems unnecessary. Latter in the paper it becomes clear that you do use it. Perhaps it would help if before this point some mention of why near surface temps are important and that they fluctuate a lot was added.*

We have changed the wording in this section to make it clearer that the 29 day moving average was applied to all the temperature data and to make clearer why the data needed to be smoothed.

"Thaw depth was calculated from the daily mean subsurface temperatures at each site by fitting a function to the temperature profile. The near surface (3 cm) temperature responds quickly to changes in the air temperature and as a result, it has fluctuations that would produce unrealistic variations in thaw depth. To correct for this, a 29 day moving average was applied to smooth the data at all depths."

*8) Page5 Line 26: The phrase, ". . .function fit to data pass through each measurement point. . ." is awkward, try to rephrase.*

Rephrased to, "…function used passed through each measurement point…"

*9) Page5 Line 31: '. . .at this site is shown. . .' Replace 'this' with KC1. Also I noticed throughout the manuscript that 'this' is used a lot, when it would help to be more specific and clearer to say what 'this' is.*

Agreed and changed accordingly.

*10) Page6 Line 17: Again 'this' in "Fovell (1997) used this approach" is vague. Did Fovell use the cluster or rule-based approach?*

Agreed, this was a little vague, 'this' was changed to "a cluster analysis"

*11) Page6 Line 20: May be helpful to reference figure 6 here for an example of a dendrogram.*

Agreed, figure referenced.

*12) Page12 Line1-6: The tussock discussion is interesting in that it details how plants and ecosystems can govern environmental conditions. My question is, wouldn't the thermal conductivity of the tussock have to be high or relatively higher than snow to able to conduct*

*energy from the subsurface to the atmosphere in order for the winter cooling affect to happen. While not within the scope of this paper, modeling schemes maybe able to define what thermal conductivities of tussocks are necessary to have a cooling effect, or what densities of tussocks sticking up above the snow are necessary.*

While the reviewer is correct, the thermal conductivity of a tussock would likely be rather low, we specifically use the word convection, not conduction, because the holes created by tussocks in the snow cover during early winter season allow warm air at the ground surface to be convectively replaced by colder and thus higher density atmospheric air.

*13) Page12 Line29-33: The discussion of the interaction between the river disturbance and plant community succession is an important result/discussion point of the paper as it provides another example of 1) the interaction between geomorphology and ecology, and 2) how plant community succession determines the physical environment (i.e subsurface temperature). I would suggested that this point be highlighted more as it could provide further evidence as to 1) why ecotype classification can be used to map permafrost conditions and 2) that understanding the interaction of disturbance and the direction of plant community succession will help inform permafrost evolution.*

We agree that this is an important discussion point in the paper and have added to this paragraph to help point out the importance of this example.

"This example underscores the tight coupling between ecotypes and ground thermal regime, which is a result of the coevolution of ecotypes, geomorphology, and ground thermal regime, rather than a causational relationship."

*14) Page13 Line 8: What do you mean by grid-based approaches? Finite difference and finite-volume or spatially distributed GCM's models come to mind. CLM and many spatially distributed models have plant functional type representation and ways of simulating ecotypes and the effects of those types on permafrost. Here I agree that models should link ecological types to the physical environment, but what is to limit grid based models from doing this?*

As both reviewers did not like the comment about grid-based approaches, it has been removed and suggested instead that the ecotype approach offers an improvement in spatial resolution without increased computational demand because the model only needs to be run once for each ecotype rather than once for every grid cell.

"Accordingly, we recommend that future permafrost modeling efforts consider using an ecotype approach as it offers increased spatial resolution without increased computational demand (i.e. a model only needs to be run once for each ecotype)."

*15) Page12 Line 25: Is it appropriate to bring up funding here? Financial constraints at some point limit most studies as has already been acknowledge on page 4 line 8, but is this publication the appropriate place to discuss the lack of money in sciences? I know Robinson R44 helicopters are expensive, despite being supper cool. However, it may be better to discuss the benefits of*

*continued and additional data gathering, which would then provide motivation for continued funding. How might more measurements build confidence in the ecotype approach and reduce uncertainty in permafrost assessment.*

The reviewer is correct, we don't need to mention funding here. This statement has been removed.